# Analysis of Mercury Content in Various Types of Tea (*Camellia sinensis*) and Yerba Mate (*Ilex paraguariensis*)

**DOI:** 10.3390/ijerph19095491

**Published:** 2022-05-01

**Authors:** Barbara Brodziak-Dopierała, Agnieszka Fischer

**Affiliations:** Department of Toxicology and Bioanalysis, Faculty of Pharmaceutical Science, Medical University of Silesia, 30 Ostrogórska Str., 41-200 Sosnowiec, Poland; afischer@sum.edu.pl

**Keywords:** tea, types of tea, Yerba Mate, mercury

## Abstract

Due to the content of active ingredients, teas can be used prophylactically, but most of all they are consumed for taste reasons. As with food or water, these products can be contaminated with heavy metals, including mercury. Mercury (Hg) is a toxic element, it causes many side effects in the human body depending on the form of Hg, which can include respiratory failure, kidney damage, neurological disorders. At the cellular level, Hg and its compounds lead to a disturbance of metabolism and cell death. The aim of the study was to evaluate the mercury concentration of tea (*Camellia sinensis*) and Yerba Mate (*Ilex paraguariensis*). Eighty-six samples were collected and analyzed, including the following kinds: black, green, white, Pu-erh, and Yerba Mate. The samples came from Poland. The Hg concentration was determined with an AMA 254 atomic absorption spectrometer. The study showed that the Hg content in each tea sample averaged 2.47 μg/kg. The Hg concentration in the tested types of tea differed significantly statistically (*p* = 0.000). It was the largest in Yerba Mate, followed by green, Pu-erh, and white tea, and was the smallest in black tea. Statistically significant differences in the Hg content (*p* = 0.004) were also dependent on the form of the product; in leaf tea samples, the concentration of Hg (2.54 µg/kg) was higher than in tea bags (1.16 µg/kg). The Hg concentration determined in the tested samples does not exceed the permitted EU standard. Consuming these teas poses no health risk in terms of the amount of Hg.

## 1. Introduction

Tea is grown and produced in over 58 countries in the world—mainly in Asia [1]. Tea is very popular and its use is constantly increasing. As an infusion, after water, it is the most consumed drink in the world. In addition, it is valued for its nutritional and health-promoting properties [2].

Environmental pollution causes toxic substances, including heavy metals, to be present in products of natural origin. The leaves contain greater amounts of mercury due to the fact that it can get there from the soil through the root system and conductive system, as well as directly from the atmosphere. Hg is collected from the soil through the root system and then accumulated in plant tissues. Heavy metals in plants can also result from the influence of air and water pollutants [3].

The tea is obtained from the Chinese tea bush (*Camellia sinensis*). The leaves, stems, and buds are used as raw materials. The types of teas differ based on the part of the plants from which they are obtained, the method of raw material processing, and the degree of fermentation. This, in turn, affects the content of active ingredients and pro-health properties. The composition of tea includes many active compounds like polyphenols (mainly catechins), proteins, amino acids, carbohydrates, lignins, fats, organic acids, carotenoids, chlorophyll, purine alkaloids (caffeine, theophylline, theobromine), and minerals, as well as vitamins, which are mainly from group B [4]. The content of active ingredients makes the tea exhibit health-promoting properties. Moreover, it can have a beneficial effect on the treatment of diseases [2,5]. Consuming tea is recommended for neurological diseases, obesity, cardiovascular disease, type 2 diabetes, and certain types of cancer (ovary, lung, skin, breast, endometrium, prostate, bladder, mouth, and colon). It has a positive effect, among others, on regulation of blood pressure, sugar levels, lowering total cholesterol in the blood, reducing body weight, or slowing down the aging process [6,7].

The most popular type of tea is black tea, and drinking it regularly helps to reduce the risk of cardiovascular diseases, including coronary artery disease, atherosclerosis and strokes by regulating blood pressure and lowering total cholesterol [8,9,10].

Green tea has the highest concentration of active compounds and the highest antioxidant properties. This tea contains catechin compounds (including epigallocatechin gallate) and polyphenols. The content of polyphenols in green tea varies from 30% to 40%, while in black tea it varies from 3% to 10% [11,12,13,14]. Green tea stimulates the central nervous system, increases resistance to stress, prevents the occurrence of neurodegenerative diseases, reduces lipid levels, and regulates the ratio of LDL (low-density lipoprotein) to HDL (low-density lipoprotein), thereby inhibiting the formation of atherosclerotic plaques. Moreover, it inhibits digestive enzymes and fat absorption more strongly than black tea, which results in the reduction of waist circumference, abdominal fat, and weight reduction, thereby preventing obesity. It also increases the sensitivity of tissues to insulin and stimulates its secretion, regulating the level of glucose in the blood. The anti-inflammatory effect is manifested by reducing protein denaturation and increasing the production of pro-inflammatory cytokines [15,16,17]. It was shown that consumption of green tea significantly reduced the risk of developing liver cancer and had a positive effect on the body mass index (BMI) and liver enzymes [18].

White tea is one of the least processed teas, the fermentation process is negligible; together with green, it is classified as unfermented tea and it is valued for the large amount of active ingredients, e.g., polyphenols, amino acids, microelements, and vitamins [19,20,21,22].

A characteristic ingredient of Pu-erh tea is theobromine. It has the strongest slimming effect of all teas. It stimulates metabolism, food digestion, reduces adipose tissue, increases bile secretion, and contributes to lowering the level of total cholesterol, LDL fraction, and triglycerides [23,24,25,26].

Among the teas, Yerba Mate is a separate type, which comes from the Paraguayan holly (*Ilex paraguariensis*). Infusions prepared with Yerba Mate have a high biological activity due to the high content of polyphenols (quercetin, rutin), phenolic acids (caffeic and chlorogenic acids), saponins, and caffeine. Yerba Mate stimulates the body, eliminates fatigue, and improves short-term memory, alertness, and reaction time, and it also contributes to weight reduction and prevents obesity [27,28,29,30].

The number of studies confirming the health benefits of tea is steadily increasing. However, the amount of tea you consume that is associated with its potential health benefits, as indicated by research, varies. It is difficult to determine the optimal amount and frequency of tea consumption to promote health [31].

Residues and contaminants in tea may pose health hazards to tea drinkers. There is a growing concern about Hg contamination in tea and infusions [2,3,31].

Poland holds a high position in the world, and in Europe, in terms of the amount of tea consumed [32]. According to statistics, the import of tea to the country in 2018 amounted to 36.4 thousand tons and the monthly consumption of tea was estimated at 50 g [7,33]. It is estimated that a citizen of Poland drinks 94 L of tea per year. Double the amounts of tea infusions, the largest among European countries, are consumed by the inhabitants of the UK and Ireland (approx. 190 L per year per person) [34].

Both environmental pollution and technological processes can affect the content of heavy metals in teas. Mercury (Hg) is highly mobile in the environment and has the ability to accumulate in ecosystems. In the body, Hg inhibits protein synthesis, disrupts the activity of neurotransmitters, and damages neurons. The neurotoxic effects of mercury interfere with cell proliferation, gene expression, cell signaling pathways, protein phosphorylation, and calcium ion homeostasis. Mercury also has a nephrotoxic effect associated with a reduction in the activity of antioxidant enzymes [35,36,37,38,39,40].

The aim of this study was to analyze the mercury content in various types of tea (*Camellia sinensis*) and Yerba Mate (*Ilex paraguariensis*). In particular, these were: leaf (selected) and tea bags with and without additives. The study conducted was aimed at determining the contamination of products with mercury and the potential risk for humans. The Hg concentration in teas was compared to the applicable EU permissible standards [41].

## 2. Material and Methods

Eighty-six different tea samples were used in the research. The characteristics of the material for the research are presented in Table 1. In terms of the quality of the dried plants, these were: leaf (selected) and express tea bags (made from smaller, crushed, pieces of leaves and buds left after selection). Most of the teas were leaf teas (60 samples), while 26 samples were tea bags. Among the collected samples, the following teas were distinguished: black, green, white, Pu-erh, and Yerba Mate. Single-ingredient teas (i.e., consisting of homogeneous dried plants with no additional ingredients in their composition) and additives teas (multi-ingredient with natural additives (fruit, flowers, fruit peels, herbs) and synthetic additive (flavors)) were used. The tested teas were purchased at various points of sale in Poland. These were general grocery stores, supermarkets, and specialist tea sales points, such astea houses. The products came from various (both domestic and foreign) manufacturers.

About 4 g of each tea intended for testing was taken and ground in a porcelain mortar. Then, three independent weights (approx. 50 mg, analytical balance RADWAG, Radom, Poland) were prepared and used for analysis.

The mercury concentration in the tea samples was determined by the AAS method (AMA 254 Mercury Analyzer, Altec, Praha, production Czech Republic). The measurement conditions used were: wavelength 253.65 nm, carrier gas-oxygen (O_2_ purity ≥ 99.5%), and inlet pressure 200–250 kPa. The times for the individual stages of the analysis were: 120, 140, and 60 s. Before each measurement, the apparatus was purged with air and distilled water according to the analytical procedure [42]. The limit of detection is 0.01 ng/g. The original factory calibration was still valid for the calibration of the instrument [42]. In order to verify the correctness of the applied method, Hg determinations were used in the reference material Mixed Polish Herbs INCT-MPH-2, Institute of Nuclear Chemistry and Technology Department of Analytical Chemistry, Warsaw, Poland. The results of the mercury determinations for six repetitions for the reference material was 0.0166 ± 0.0001 mg/kg with a recovery of 92.22%. The detection technique used covers the total amount of Hg, regardless of its form in the sample. The final Hg concentration in the tested tea sample was the arithmetic mean of three measurements. The statistical analysis of the obtained results was performed with the use of Microsoft Excel and Statistica 13.3 programs (Statsoft, Cracow, Poland). The distribution of the variables was evaluated by the Shapiro–Wilk test and quantile–quantile plot. The interval data were expressed as a median (lower-upper quartiles). The nonparametric Mann–Whitney U test (for two samples) and Kruskal–Wallis test (for a greater number of samples) were used in order to compare the data. The statistical significance was set at a *p*-value below 0.05 and all tests were two-tailed [43].

## 3. Results

The statistical analysis of the Hg concentration in the tested tea samples is presented in Table 1. The average Hg concentration in all the tested samples was 2.47 µg/kg and the range of changes was 0.36–10.76 µg/kg. The concentration of Hg in the individual types of test samples showed a statistically significant difference (*p* < 0.001)—shown in Table 1. A careful analysis of the intergroup probability revealed statistically significant differences in the Hg content in Yerba Mate and the other tea plant (*Camellia sinensis*) species tested (*p* < 0.001). The Hg concentration in the Yerba Mate samples was 5.67 µg/kg, and in all the other tested samples there was a total of 2.51 µg/kg. Among the tea samples, the highest concentration of Hg (but more than two times lower than in Yerba Mate) was found in green tea (2.47 µg/kg), followed by Pu-erh and white tea—shown in Figure 1. The average concentration of Hg was the lowest in the samples of black tea (1.21 µg/kg)—shown in Table 1.

When comparing the form of dried plants, it was shown that for all the tested samples, the leaf form contained a higher concentration of Hg (2.54 µg/kg) than the tea bag form (1.16 µg/kg)—the differences were statistically significant (*p* < 0.01), as shown in Figure 2. 

The analysis of the Hg concentration in the individual types of leaves and express tea (tea bags) is presented in Figure 3. A higher Hg concentration in the leaf form was found in black, green, and white teas, with statistically significant differences only being seen in black tea (*p* = 0.007).

The Pu-erh and Yerba Mate teas analyzed in the study were found only in the leaf form, so the above comparison was not used. On the other hand, when comparing the Hg concentration in all the tested samples of pure (single-ingredient, no additives) and complex teas (containing other ingredients in addition to dried tea), it was observed that the additional components did not significantly affect the Hg concentration (*p* > 0.05). The average level of Hg in pure tea samples was 2.86 µg/kg and in samples with additives it was 2.31 µg/kg—shown in Figure 4.

Taking into account the type of tea in this comparison, statistically significant differences occurred only in the case of the tested green tea samples (2.55 µgHg/kg—tea with additives and 4.09 µgHg/kg—pure tea). Similarly, in the case of Yerba Mate, additional substances supplementing the composition significantly (*p* = 0.002) influenced the increase in the Hg concentration in the tested samples (8.28 vs. 5.14 µgHg/kg)—Figure 5.

Taking into account the type of components supplementing dried tea, it was shown that in the case of the additive of natural origin (flower, fruit, fruit peel, herbs), the average Hg concentration in the finished, tested product was higher than in the case of the addition of synthetic flavors (aroma)—shown in Figure 6 (statistically significant differences, *p* < 0.01). Among the tea samples tested, the highest Hg concentration was found in mixtures with natural additives in the form of fruit peels, herbs, whole fruits, and flowers.

Moreover, the conducted study did not show any statistically significant differences in the concentration of Hg in tea samples purchased at various points of sale. A comparable level of Hg content was found in products purchased in specialized tea houses (classified as high-quality/more expensive) and in grocery stores or supermarkets.

## 4. Discussion

The tea bush (*Camellia sinensis*) and the Paraguay Holly (*Ilex paraguariensis*) may contain heavy metals, the concentration of which may vary depending on the country of origin of the plant [34,44,45] and the quality and composition of the soil on which it grows [45,46,47,48]. The raw materials are obtained from the above-ground parts of plants, which undergo technological processes that may also affect the content of pollutants. Consuming tea is considered a good source of nutrients for the body. These drinks are recommended for supporting the physiological functions of the body, being used prophylactically in the prevention of diseases, and sometimes also as therapeutic agents [6]. However, these consumed infusions can lead to the exposure of the body to toxic components [49]. National and European regulations indicate, though considered safe, the levels of toxic substances in products consumed by humans. According to EFSA, the TWI (Tolerable Weekly Intake) value for methylmercury is 1.3 μg/kg b.w. and 4 μg/kg b.w. for inorganic mercury [50]. For the residual Hg compounds in teas, as well as coffee beans, cocoa beans, and herbal infusions, an acceptable limit of 0.02 mg/kg was adopted [41]. The analysis carried out in the research showed that none of the tea samples tested exceeded the prescribed standards. The average Hg concentration for all the tested samples was 2.86 µg/kg (range of change: 0.36–10.76 µg/kg). The lowest concentration of Hg was found in the samples of black tea (1.63 µg/kg) and the highest was found in Yerba Mate (6.87 µg/kg). Research on the mercury content in teas is scarce. Moreover, the results of research in this area are highly diversified. Gajewska et al. [51] determined the content of minerals and toxic components in black and green tea. The range of Hg content in these tests was 2.5–15.8 µg/100 g [51] and it was many times greater than in our determinations (0.036–1.07 µg Hg/100 g). When comparing the type of tea, the concentration of Hg in the green tea samples, shown in the above studies, was 3.9 µg/100 g [51], which was about 10 times higher than indicated by our analysis (0.326 µg/100 g). On the other hand, the concentration of Hg in the green tea samples tested by Melluci et al. [52] was determined to be in the range of 0.23–0.69 µg/g, which are higher values than we found in our research. Both studies by Gajewska et al. [51] and Schulzki et al. [34] showed that the concentration of Hg was higher in black tea than in green tea. In the case of our study, an inverse was observed, where green tea samples had a higher amount of mercury compared to black tea (3.26 vs. 1.63 µg/kg).

In some countries, drinking tea is a daily ritual and tradition. In Iran, tea consumption among the inhabitants is particularly popular and amounts to around 1.3 kg/year/person [53]. The Hg concentration in teas available on the market of this country was presented in the review article by Hashemour-Baltrok et al. [54]. In the most popular black teas among consumers, both domestic and imported, which were tested by Falahi and Hedaiati [55], the concentration of Hg ranged from 0.01–0.20 mg/kg and averaged 0.13 mg/kg [52]. In other studies in this area, the average Hg concentration was found to be 0.61 mg/kg [56]. In our research of tea samples, the range of the Hg concentration was 0.36–10.76 µg/kg.

In the teas available on the retail market in Poland, the Hg concentration was tested by Kolwalski et al. [56]. The same research apparatus that was used in our research showed a comparable level of Hg concentration in the tea samples. For black tea, the Hg concentration was in the range of 0.001–0.009 mg/kg [56], while in our research the average Hg concentration in this type of tea was 0.002 mg/kg. As in our research, the concentration of Hg in green tea was higher than in black tea. For the Pu-erh tea samples tested, the Hg concentration was 0.0024 mg/kg [56] and, as in our research, this level was comparable to the Hg content in green tea. However, in the research conducted by Li et al. [57] in raw Pu-erh tea, the Hg concentration was 0.01–0.03 mg/kg. It was also shown that over the years (2005–2019) the concentration of metal in the tested samples was systematically decreasing, but it had no statistical significance (*p* = 0.5) [57].

Among the samples tested by us, the highest concentration of Hg was found in Yerba Mate. This type is most often the subject of analysis separate from those of tea [27,29,30]. Moreover, data from the literature show that the content of both micro- and macro-elements and toxic elements in Yerba Mate is higher than in teas [58,59,60]. In our research, the Hg concentration in Yerba Mate was 6.87 µg/kg, which was several times higher than in black (1.63 µg/kg) and green (3.26 µg/kg) tea. Proch et al. [45] tested Yerba Mate samples from different countries (Argentina, Paraguay, and Brazil), both in pure form and with additives. The Hg concentration in these samples was 0.03–1.34 mg/kg with a mean of 0.30 mg/kg [45]. It was shown that the concentration of Hg in the samples containing additives was higher (0.37 mg/kg) than in the case of pure products (0.29 mg/kg) [45]. We observed a similar thing in our research. In the case of Yerba Mate samples containing additives, the concentration of Hg was greater than in the pure form (7.86 vs. 5.09 µg/kg). The differences in the content of Hg in these types of samples were statistically significant. A similar, although statistically insignificant, tendency was observed in relation to black tea. However, the analysis of green tea samples, in which the average level of Hg content was higher in the pure form than in the form with additives, was shown differently. The results of our research indicated that the concentration of Hg was higher in the test samples containing additives in the natural form (e.g., peels of fruit, herbs, whole fruits, and flowers) than the flavored products, which is most likely due to the presence of environmental pollutants, including Hg, in plant products. Among the analyzed natural additives, the highest concentration of Hg was found in samples containing herbs, followed by fruits and parts of spice plants. In the studies by Kowalski et al. [53], the highest concentration of Hg was found in mono-fruit tea (0.031 mg/kg), which our research on fruit teas did not include.

The usable form of tea in the form of leaves is considered to be a better product than the form sold in tea bags. Tea bags contain small, poorer quality plant particles. Our research on various tea samples showed that both white, green, and black leaf teas contained significantly higher concentrations of Hg than in the express tea bags. The research of Wojciechowska-Mazurek et al. [61] showed similar results, where the average concentration of Hg in leaf tea was higher than in tea bags or granulated tea. The maximum values obtained for the samples of different qualities indicate the influence of the technological process on the concentration of heavy metals in tea products.

Most of the levels of elements found in the literature for teas relate to dried plant raw materials. Both nutritional and toxic substances contained in plants used to prepare infusions go into solution [34,61,62]. A study by Schulzki et al. [34] shows that the transition of the elements to infusions may vary. For example, the transition rate for Pb is 25%, while for Cu it is 50%. In the case of elements with a low concentration (below the detection level) in dried plants, such as Cd or Hg, we can determine theoretical transition rates [34]. The popularity of tea drinks means that consuming large amounts of infusions, even with a relatively low content of metals, increases the amount of consumed ingredients. In the studies by Colapinto et al. [60], it has been shown that in the blood of pregnant women who regularly consumed green tea, the concentration of mercury and arsenic was significantly higher than those who did not drink tea. There is also no generally accepted rule on how to prepare infusions. They can be prepared in different ways, e.g., varying the dry product to water ratio or using different temperatures or brewing times. All these factors contribute to the different individual exposures. Research by Wang et al. [63] showed that the method of preparing a green tea infusion has an impact on the value of the leaching coefficient for Hg. A clear increase in its value was observed when the brewing temperature reached 97 °C.

With the relatively high popularity of tea drinks, especially among some populations, this source of exposure to Hg for the human body cannot be ignored.

## 5. Conclusions

Mercury was found in all the analyzed samples of tea and Yerba Mate. The highest average concentration of Hg was found in Yerba Mate. The Hg content in these samples was about twice as high as in all tested tea samples.

The average concentration of Hg in teas depended on its type, with the highest being in green tea, followed by Pu-erh, white, and black tea. There were statistically significant differences in the amount of Hg depending on the form of tea (leaf vs. bags) and the presence or absence of additives in tea.

Consuming the tested tea samples, in light of the applicable legal norms, does not pose a health risk in terms of the amount of Hg. However, as a result of increasing environmental pollution with heavy metals, mercury content in plant-based foods should be monitored to ensure consumer safety.

## Figures and Tables

**Figure 1 ijerph-19-05491-f001:**
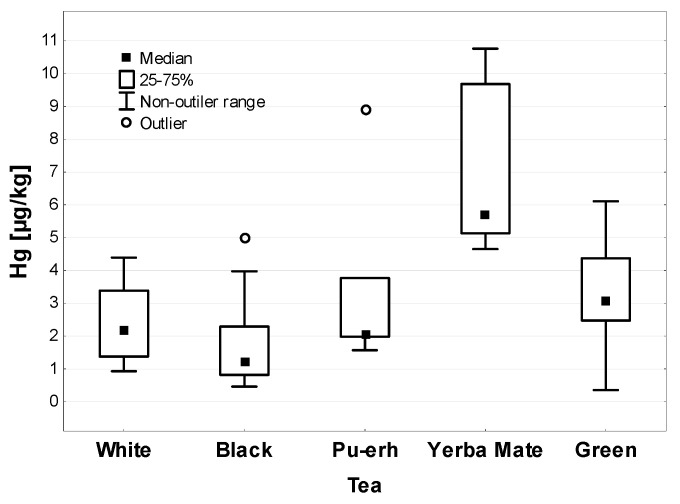
Hg concentration in different kinds of tea (µg/kg).

**Figure 2 ijerph-19-05491-f002:**
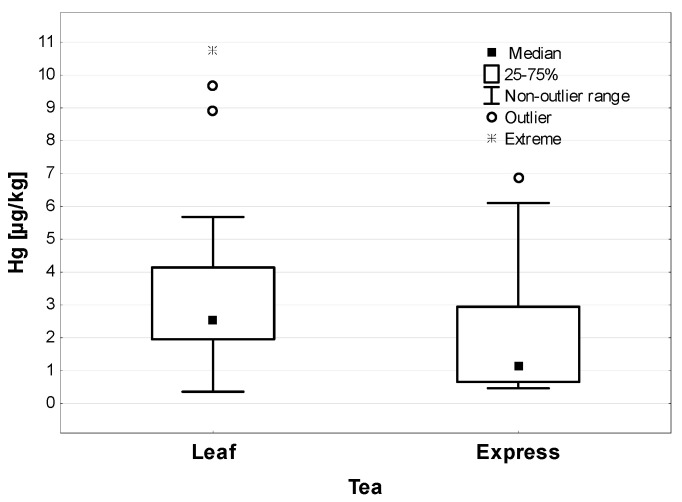
Hg concentration in leaf and express tea (µg/kg).

**Figure 3 ijerph-19-05491-f003:**
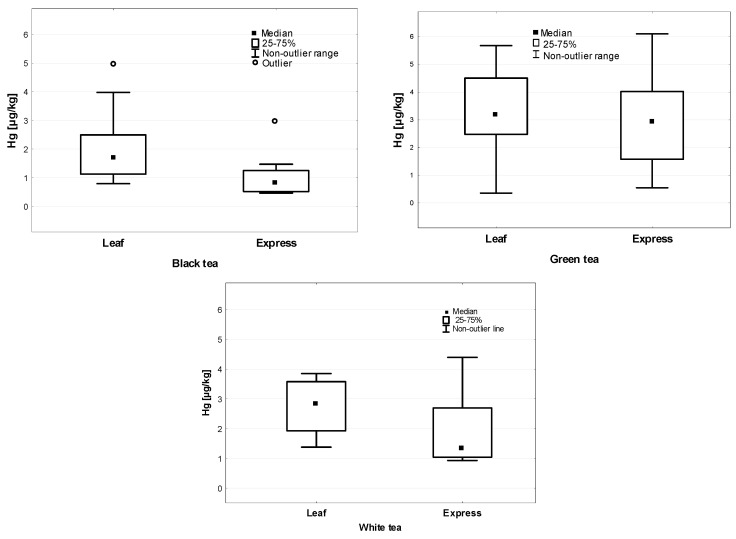
Hg concentration in different kinds of leaf and express tea (µg/kg).

**Figure 4 ijerph-19-05491-f004:**
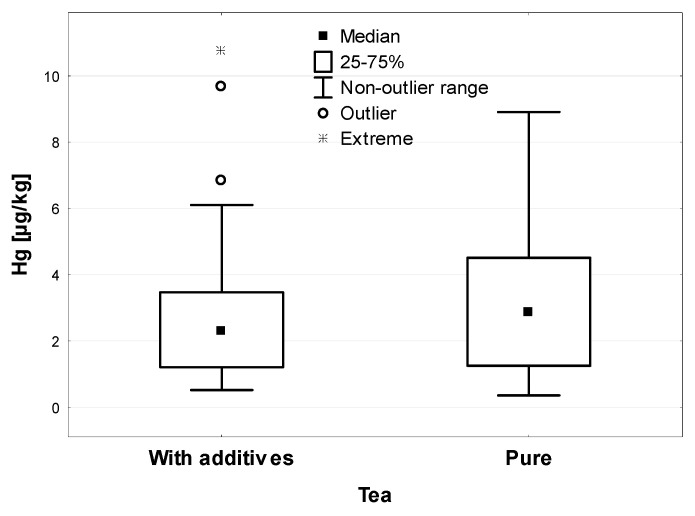
Hg concentration in pure and additives tea (µg/kg).

**Figure 5 ijerph-19-05491-f005:**
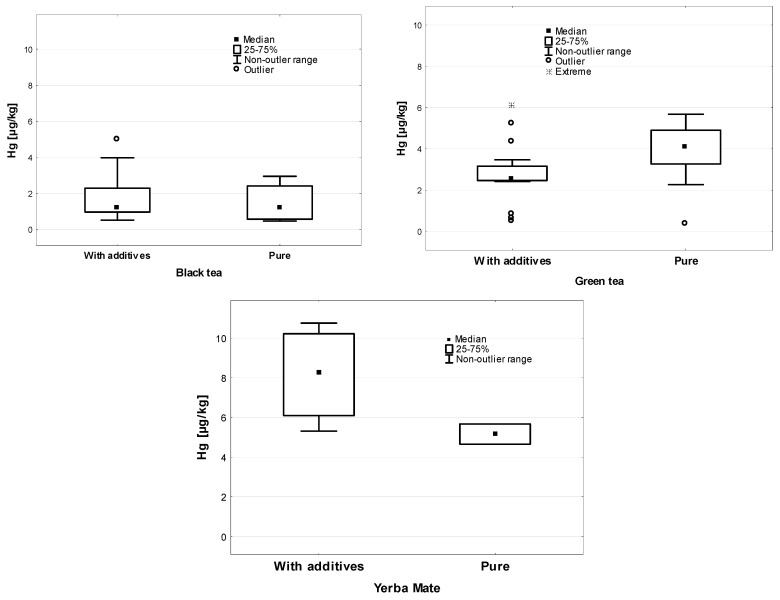
Hg concentration in different kinds of pure and with additives tea (µg/kg).

**Figure 6 ijerph-19-05491-f006:**
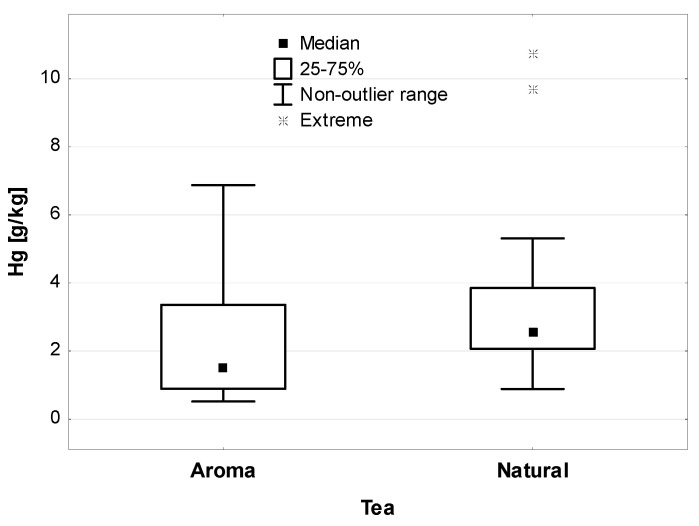
Hg concentration in tea with additives of natural origin and synthetic aroma, (µg/kg).

**Table 1 ijerph-19-05491-t001:** The concentration of Hg in tea samples [µg/kg].

Tea		N	AM ± SD	Median	Quartile		*p*
					Q_1_	Q_3_	
All		86	2.86 ± 2.07	2.47	1.21	3.88	
Kind	Black	29	1.63 ± 1.16	1.21	0.82	2.29	<0.001
	White	14	2.40 ± 1.18	2.21	1.38	3.39	
	Pu-erh	7	3.22 ± 2.60	2.06	1.99	3.77	
	Green	29	3.26 ± 1.54	3.06	2.47	4.37	
	Yerba mate	7	6.87 ± 2.41	5.67	5.14	9.68	
Form	Leaf	60	3.21 ± 2.09	2.54	1.95	4.14	<0.01
	Express	26	2.05 ± 1.80	1.16	0.66	2.94	
Composition	With additives	53	2.75 ± 2.11	2.31	1.21	3.47	0.429
	Pure	33	3.04 ± 2.02	2.86	1.25	4.51	
Additives	Synthetic aromat	19	2.32 ± 1.98	1.52	0.89	3.36	<0.001
	Natural	34	3.23 ± 2.22	2.52	2.06	3.85	

AM—arithmetic mean, SD—standard deviation, Q_1_—first quartile, Q_3_—third quartile.

## Data Availability

Not applicable.

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
