# Peer review of "Analysis of Mercury Content in Various Types of Tea (*Camellia sinensis*) and Yerba Mate (*Ilex paraguariensis*)"

_ijerph, 2022, doi:10.3390/ijerph19095491_

Round 1
Reviewer 1 Report
The article“Analysis of Mercury Content in Various Types of Tea ” is to evaluate the mercury concentration in tea. 86 samples were collected and analyzed. There are certain problems in the article, which still need to be revised. It is recommended to major revision. The specific problems are as follows:
- Since the article is to analyze the mercury contained in tea, where does the mercury in tea come from, or what is the way of entering it, it should be supplemented.
- There are many abbreviations in the text, but their full names do not appear. Please check and modify the full text.
- The use of various methods is mentioned many times in the text, and no references are cited nor introduced in the text.
- The innovation of this paper and the significance of this study are not outstanding enough, and I hope to provide additional explanations.
- Since we are studying tea, and the main way of eating tea is by brewing, why not discuss the mercury content in liquid tea?
- The discussion part of the article in this article does not revolve around the data in the article, please discuss according to the data in the article.
- It is recommended to compare the data collected in this paper with the standards for mercury content in various countries and regions.
Author Response
- the linguistic quality of the paper – We inform that the manuscript has been verified for linguistic correctness (certificate attached)
- Environmental pollution causes that toxic substances, including heavy metals, may be present in products of natural origin. Hg is collected from the soil through the root system and then accumulated in plant tissues. Heavy metals in plants can also result from the influence of air and water pollutants
- The abbreviations are explained in the text of the manuscripy
- The abbreviation of the determination method has been clarified, literature sources for the determination method are [42] and for statistical methods [43]
4, 5 Most of the levels of elements found in the literature for teas relate to dried plant raw materials. Both nutritional and toxic substances contained in plants used to prepare infusions go into solution [34, 61, 62]. A study by Schulzki et al. [34] shows that the transition of the elements to infusions may vary. For example, the transition rate for Pb is 25% and for Cu 50%. In the case of elements with a low concentration (below the detection level) in dried plants, such as Cd or Hg, we can determine theoretical transition rates [34]. The popularity of tea drinks means that consuming large amounts of infusions, even with a relatively low content of metals, increases the amount of consumed ingredients. In the studies by Colapinto et al. [63] it has been shown that in the blood of pregnant women who regularly consumed green tea, the concentration of mercury and arsenic was statistically significantly higher than in those who did not drink tea. There is also no generally accepted rule on how to prepare infusions. They can be prepared in different ways, e.g. varying dry product to water ratio, different temperature or brewing time. All these factors contribute to the different individual exposure. Research by Wang et. al [64] show that the method of preparing green tea infusion has an impact on the value of the leaching coefficient Hg. A clear increase in its value was observed when the brewing temperature reached 97°C.
- The discussion has been changed
- The data has been compared with the standards for mercury content in various countries.
Reviewer 2 Report
86 tea samples including 55 leaf teas and 28 tea bags “(55+28=83)?” in types of black, green, white, Pu-erh and Yerba Mate from general grocery stores, hypermarkets and specialist tea sales points-tea houses were applied to evaluate the mercury concentration in tea in this work. Regardless the methodology, in my opinion, the conclusion that “for none of the tea samples tested, the prescribed standards for Hg concentration were not (WRONG ? no not?) exceeded” is predictable as all the samples were commercial products which must meet certain mandatory standards.
The results of this work might be useful for administrative authorities as reference in market surveillance, but not a research article attributes.
Author Response
There was a confusion in the number of 86 samples (60 leaf teas and 26 tea bags).
The indicated conclusion has been deleted.
Reviewer 3 Report
The authors are thanked for writing clearly, it is an easy to understand article. And only few changes are recommended.
Introduction. Please include some results about what other authors have found, here some suggestions:
- The concentration and health risk of potentially toxic elements in black and green tea—both bagged and loose-leaf
Doi: https://doi.org/10.15586/qas.v12i3.761
- Green Tea Increases the Concentration of Total Mercury in the Blood of Rats following an Oral Fish Tissue Bolus
Doi: https://doi.org/10.1155/2015/320936
- Is there a relationship between tea intake and maternal whole blood heavy metal concentrations?
Doi: https://doi.org/10.1038/jes.2015.86
- Concentrations, leachability, and health risks of mercury in green tea from major production areas in China
Doi: https://doi.org/10.1016/j.ecoenv.2022.113279
Results.
Lines 145-147. The paragraph is confused.
Why is it that higher amounts of mercury are found in the leaves?
Author Response
The suggested references has been implemented [3, 40, 63, 64]
Heshmati, A.; Mehri, F.; Karami-Momtaz, J.; Mousavi Khaneghah, A. The concentration and health risk of potentially toxic elements in black and green tea—both bagged and loose-leaf. Quality Assurance and Safety of Crops & Foods. 2020; 12 (3): 140-150.
Janle, E.M.; Freiser, H.; Manganais, Ch.; Chen, T.Y.; Craig, B.A.; Santerre, Ch.R. Green tea increases the concentration of total mercury in the blood of rats following an oral fish tissue bolus. BioMed Res. Internat. 2015 15, ID 320936, 6. http://dx.doi.org/10.1155/2015/320936
Colapinto, C.K.; Arbuckle, T.E.; Dubois, L.; Fraser, W. Is there a relationship between tea intake and maternal whole blood heavy metal concentrations? J. Expo. Sci. Environ. Epidemiol. 2016, 26(5), 503-9. Doi: 10.1038/jes.2015.86
Wang, Q.; Wang, D.; Li, Z.; Wang, Y.; Yang, Y.; Liu, M.; Li, D.; Sun, G.; Zeng, B. Concentrations, leachability, and health risks of mercury in green tea from major production areas in China Ecotoxicol. Environ. Saf. 2022, 232, 113279.
Results.
Lines 145-147. The paragraph has been changed.
The leaves contain greater amounts of mercury due to the fact that it can get there from the soil through the root system and conductive system, and directly from the atmosphere.
Reviewer 4 Report
The manuscript presents an interesting assessment of the Hg contamination in infusions, not only tea.
In order to improve the quality of the manuscript some minor changes are suggested.
First of all, the title just includes "tea" and does not mention "yerba Mate". If authors could refers to tea and yearba mate or infusions or Camelia Sinensis and Yerba Mate the title would be more accurate.
Furthermore, yerba mate does not show either in the key words and that would difficult the future citation of this paper about yerba mate by other authors.
I would say that the manuscript is more an assessment than an analysis as mentioned in the title. Author should consider this change.
Abstract:
Line 9: delete "can"
Line 10: add "toxicity" after mercury
line 12: add: "various types" before tea
Line 13: add information about the sampling site (country, region)
Line 16: "the study showed an average mercury content of 2.47.."
Line 22: add a ponit after standard and start a new sentence at "Consuming these..."
Line 22: specify which standard: European?, EFSA standard??
Line 23 and 24: delete the last sentence.
Introduction:
line 43: add a cross reference for all these toxic effects
line 77: delete "you" and rephrese the sentence
Lines 93 and 97 must be revised and a joint description of the objectives should be included. Last sentence is more a result than and objective so delete "which made it...them". Explain which standards (european, efsa, which?).
Line 94: "studies" in plural or singular?
Line 105 refer to multi-ingredient teas but later along the manuscript authors refer more to additives (example line 170, figure 4 and 5 etc, line 188, 299 etc ). I believe it is more accurate to refer to ingredients than to additives.
Lines 212-213: why don't authors include the European limits set by EFSA and include the EFSA reference?
Line 272-273: explian why author mention her in the discussion that "selected leaf parts has been separated". Why was not mentioned in the methodology?. Does these affect the results?
Is there not a Tolerable Weekly Intake for Hg in Europe to be discussed?
Author Response
I would like to thank the Reviewers for the assessment of our manuscript, submitted comments
and indicated errors
In response to the review I would like to inform:
Reviewer #4:
The title and keywords have been changed.
The suggested changes to abstract and introduction have been made.
Abstract:
Line 9: the word "can" has been deleted
Line 10: the word "toxicity" has been added
line 12: the word "various types" has been added
Line 13: The samples came from Poland.
Line 16: The study showed an average Hg content in the tea samples was 2.47 μg/kg.
Line 22: point added after standard and start a new sentence at "Consuming these..."
Line 22: standard European
Line 23 and 24: the last sentence has been deleted.
Introduction:
line 43: cross reference for all these toxic effects has been added
line 77: the word "you" has been added
Lines 93 and 97 standard European Union
Line 94: study
Line 105 The content of additives may influence the mercury content.
Lines 212-213: the EFSA limits and reference [50] has been added
Line 272-273: The standards we refer to in the work relate to the European Union, they concern the permissible mercury concentration in teas. The TWI for methylmercury is 1.3 µg/kg bw and 4 µg/kg bw for inorganic mercury [50]. The TWI calculation is important for infusions and not for teas themselves.
We hope that the explanations and changes which were made will prove satisfactory for the Reviewers and will allow accepting the manuscript for publication.
Round 2
Reviewer 1 Report
The article“Analysis of Mercury Content in Various Types of Tea ” is to evaluate the mercury concentration in tea. 86 samples were collected and analyzed. There are certain problems in the article, which still need to be revised. It is recommended to major revision. The specific problems are as follows:
- Since the article is to analyze the mercury contained in tea, where does the mercury in tea come from, or what is the way of entering it, it should be supplemented in the manuscript to make it smoother.
- It is hoped that the author will add innovations to the Introduction section of the article.
- Why choose different types of tea when identifying different indicators in the experiment? It is suggested that the types of teas selected for comparison should be consistent for all indicators.
- It is suggested here to compare the method used to detect Hg in this study with similar literature reports to highlight the advantages, preferably in a tabular form.
Author Response
I would like to thank the Reviewers for the assessment of our manuscript, submitted comments
and indicated errors
In response to the review I would like to inform:
Reviewer #1:
- Environmental pollution causes that toxic substances, including heavy metals, may be present in products of natural origin. The leaves contain greater amounts of mercury due to the fact that it can get there from the soil through the root system and conductive system, and directly from the atmosphere. Hg is collected from the soil through the root system and then accumulated in plant tissues. Heavy metals in plants can also result from the influence of air and water pollutants [3].
- The section “Introduction” was supplemented with new data from the literature [2, 3, 31]
- Literature data [51-53, 61, 63] show differences in mercury content in different types of tea, e.g. in green teas content of Hg is higher than in black teas. The choice of different types of teas for research was caused by the high popularity of their use as a drink. As an infusion, tea after water, it is the most consumed drink in the world. Black tea is the most common type of tea, but various types are also very popular, too. The individual preferences in choosing the type of tea are different. The amounts of tea in the form of infusion also vary, but it is known that heavy metals can go into solution [34, 61, 64] and then to the human organism.
- Literature studies show that the concentration of Hg in the tea samples tested by different authors may vary significantly. As we indicate in Discussion, this may be due to the different products available to the consumers in markets. The differences may also result using the different research methods. Therefore, in the Discussion, we referred to the results of the research by Kowalski et al. [56], who used the same research technique we in our research (AAS method, Mercury Analyzer AMA 254, Altec, production Czech Republic) and tested tea samples available in the same market (Poland). This aspect is indicated in the Discussion and described. We have doubts whether the compilation of the results in the form of a table without any additional explanations (which we have provided above) will be sufficiently legible by the recipient.
We hope that the explanations and changes which were made will prove satisfactory for the Reviewers and will allow accepting the manuscript for publication.
Best regards,
Authors of the manuscript
Reviewer 2 Report
This work was a survey report of significance in market surveillance and environmental monitoring, other than a scientific research paper. As was pointed out in the previous review that the results were predictable as all the samples were commercial products which must meet certain mandatory standards.
Author Response
Reviewer #2:
I would like to thank the Reviewers for the assessment of our manuscript, submitted comments
and indicated errors
In response to the review I would like to inform:
All the samples in this our study were commercial product. They comes from various points of sale in Poland. As a commercial product, they should meet certain standards. Numerous literature reports indicate that teas, as a commercial product, are the subject of the research by various authors and concern markets in different countries [2, 4, 47-49, 51-56]. They are especially popular where drinking tea is a national tradition. The results of these studies are published. On the basis of the publications, it can be indicated, for example, whether the content of elements in particular types of tea is different. It is known that elements, including heavy metals, only partially pass into the infusion during the preparation of the drink. However, with a large individual consumption, the amount of elements absorbed into the body is higher. For example, in the studies by Colapinto et al. [63] it has been shown that in the blood of pregnant women who regularly consumed green tea, the concentration of mercury and arsenic was statistically significantly higher than in those who did not drink tea. The assumption of our experimental research was to confirm the level of Hg content in various teas on the Polish market. We hope that other authors may find such indications interesting and they may be used for future discussions.
The determination of the Hg content in tea samples was conducted by us as author's research.
We hope that the explanations and changes which were made will prove satisfactory for the Reviewers and will allow accepting the manuscript for publication.
Best regards,
Authors of the manuscript
